# Psychosis Caused by a Somatic Condition: How to Make the Diagnosis? A Systematic Literature Review

**DOI:** 10.3390/children10091439

**Published:** 2023-08-23

**Authors:** Nolwenn Dissaux, Pierre Neyme, Deok-Hee Kim-Dufor, Nathalie Lavenne-Collot, Jonathan J. Marsh, Sofian Berrouiguet, Michel Walter, Christophe Lemey

**Affiliations:** 1Centre Hospitalier Régional et Universitaire de Brest, 2 Avenue Foch, 29200 Brest, France; 2Unité de Recherche EA 7479 SPURBO, Université de Bretagne Occidentale, 29200 Brest, France; 3Fondation du Bon Sauveur d’Alby, 30 Avenue du Colonel Teyssier, 81000 Albi, France; 4Laboratoire du Traitement de l’Information Médicale, Inserm U1101, 29200 Brest, France; 5Graduate School of Social Service, Fordham University, 113 West 60th Street, New York, NY 10023, USA

**Keywords:** first-episode psychosis, differential diagnosis, initial work-up, systematic review

## Abstract

Background: First episode of psychosis (FEP) is a clinical condition that usually occurs during adolescence or early adulthood and is often a sign of a future psychiatric disease. However, these symptoms are not specific, and psychosis can be caused by a physical disease in at least 5% of cases. Timely detection of these diseases, the first signs of which may appear in childhood, is of particular importance, as a curable treatment exists in most cases. However, there is no consensus in academic societies to offer recommendations for a comprehensive medical assessment to eliminate somatic causes. Methods: We conducted a systematic literature search using a two-fold research strategy to: (1) identify physical diseases that can be differentially diagnosed for psychosis; and (2) determine the paraclinical exams allowing us to exclude these pathologies. Results: We identified 85 articles describing the autoimmune, metabolic, neurologic, infectious, and genetic differential diagnoses of psychosis. Clinical presentations are described, and a complete list of laboratory and imaging features required to identify and confirm these diseases is provided. Conclusion: This systematic review shows that most differential diagnoses of psychosis should be considered in the case of a FEP and could be identified by providing a systematic checkup with a laboratory test that includes ammonemia, antinuclear and anti-NMDA antibodies, and HIV testing; brain magnetic resonance imaging and lumbar puncture should be considered according to the clinical presentation. Genetic research could be of interest to patients presenting with physical or developmental symptoms associated with psychiatric manifestations.

## 1. Introduction

First episode of psychosis (FEP) is a full-threshold clinical condition in which altered perceptions of reality, delusions, disorganized thoughts, and cognitive dysfunctions can be observed by others and are associated with functional decline [1]. FEP affects 3% of the population and typically manifests during adolescence or early adulthood. Additionally, FEP may pose severe consequences, such as suicidal behaviors. Following FEP, the clinical trajectory leads to a diagnosis of schizophrenia in 51% of cases and 32.5% for another non-affective psychotic disorder [2]. However, psychotic-like symptoms have been found to also reveal a somatic affection; for example, it is well-established that auditory hallucinations can be caused by temporal epilepsy [3]. Furthermore, several studies have found that psychotic-like experiences may be associated with physical disorders in ultra-high-risk patients [4] and that at least 5% of FEP could be caused by a physical disease [5,6]. It is thus essential to identify whether these somatic diseases potentially underlie FEP patients, especially since curative treatments for them exist in most cases. Moreover, over the past decade, studies have shown evidence that early detection of the first signs of psychosis and early intervention with a reduced duration of untreated psychosis (DUP) limit the negative impacts of the disorder [7,8].

Considering these results, guidelines focus on early intervention and treatment of FEP. However, the low specificity of FEP symptoms makes detection highly challenging, and even though guidelines propose a “general medical assessment” [1], no consensus of recommendation has been established for a comprehensive medical assessment to eliminate somatic causes. One of the main challenges in the case of FEP is to dual-diagnose co-occurring somatic diseases. Some pathologies may not be immediately clinically apparent, although a brief test can make the diagnosis. More commonly, there is no specific test, and the clinician will turn to a differential diagnosis based on unstructured clinical and paraclinical inferences. Furthermore, antipsychotics should be used with caution in some clinical conditions, such as encephalitis [9,10,11], indicating the sensitivity of FEP cases as well as the lethal risks that ill-informed treatment or medical mismanagement may pose. This systematic review aims to determine the physical diseases that can engender psychotic symptoms and the paraclinical exams that allow for the exclusion of these pathologies.

## 2. Materials and Methods

This review followed the Preferred Reporting Items for Systematic Reviews and Meta-Analyses (PRISMA) guidelines [12] to ensure comprehensive and transparent reporting of methods and results. The protocol was registered at the International Prospective Register of Systematic Reviews (PROSPERO) in June 2019 (registration number: CRD42020136243).

### 2.1. Search Strategy

Two independent authors searched the electronic database PubMed. A two-fold research strategy was used to ensure a complete search. The research terms used were as follows: “Schizophrenia/diagnosis”[MAJR] AND “Diagnosis, Differential”[MeSH Terms] AND review” and psychosis[Title/Abstract] AND diagnosis[Title/Abstract] AND neurologic[Title/Abstract] OR endocrinologic[Title/Abstract] OR metabolic[Title/Abstract] OR infectious[Title/Abstract] OR inflammatory[Title/Abstract] OR autoimmune[Title/Abstract] OR chromosomic[Title/Abstract] OR genetic[Title/Abstract] OR toxic[Title/Abstract].

In addition, a manual literature search was performed on the websites of academic societies to identify publications related to the subject.

Papers published until July 2021 were included.

### 2.2. Eligibility Criteria

Articles must have included the prerequisite topics, “first-episode psychosis differential diagnosis and/or related paraclinical features”, with the language of the publication written in English.

Case reports were excluded to avoid non-relevant data or duplicates with reviews already reporting matching cases. The psychotic presentation of medical conditions such as brain tumors, traumatic brain disorders, substance abuse, and common thyroid disorders has been clearly established for several decades [1,13,14]. These somatic conditions are thus not detailed in this review, as they have not been the subject of active research within the last 15 years.

### 2.3. Data Collection

All potential studies were first exported into a reference citation manager (Zotero), and duplicates were removed. Two independent researchers separately performed the screening of titles and abstracts, full-text analysis, and selection of paraclinical exams. Disagreements were resolved through discussions until a consensus was reached and a third reviewer was available to resolve the rest of the disagreements.

Next, a narrative-descriptive summary was created for the selected studies.

As the selected studies use heterogeneous methods, we identified commonalities from them through a two-part analysis: (1) by study features (number of studies, theme, populations, bias, etc.); and (2) by studies’ contributions to clinical psychiatry and implications for future research. Studies were classified according to the nosographic category they addressed (neurology, infectious diseases, etc.), the type of exam performed, and the type of study that was conducted (meta-analysis, literature review, etc.).

## 3. Results

### 3.1. Search Results

The electronic search returned 970 articles, and five recommendations were also identified through the hand-searched literature of publications in academic societies. A full-text evaluation was conducted for 130 articles, out of which 87 met the inclusion criteria and were included in the final synthesis (details in Appendix A). From these studies, autoimmune, metabolic, neurologic, infectious, and genetic differential diagnoses were identified for FEP, and the complete list of laboratory or imaging features required to confirm these diseases was drawn (detail in Appendix B).

Details of the search results are summarized in Figure 1.

### 3.2. Autoimmune Diseases

Of the 87 articles, 38 concerned autoimmune diseases, of which systemic lupus erythematosus and limbic encephalitis were the most frequently described.

#### 3.2.1. Limbic Encephalitis

Since the first description of limbic encephalitis in 1960, several antibodies responsible for these autoimmune brain inflammations have been discovered. The most frequent kind of limbic encephalitis is anti-N-Methyl-D-Aspartate (NMDA) receptor encephalitis, which predominantly affects children (35–40%) [9,15]. Other antibodies that are found in limbic encephalitis include anti-LG1, anti-GAD, anti-VGKC-Ab, anti-AMPA-r, and anti-GABA-r. Encephalitis with anti-NMDA-r, anti-AMPA-r, and anti-GABA-r antibodies has been strongly identified as being responsible for the manifestation of psychiatric symptoms [16,17,18,19,20]. It appears that psychotic symptoms are more frequent in limbic encephalitis with anti-NMDA-r antibodies in the form of drastic personality change [9], abnormal behavior (agitation, aggression, or catatonia), and delusions and hallucinations [15,21]. These symptoms can also be associated with irritability, insomnia, mood troubles [22,23], and a high level of anxiety [10,24]. Anti-NMDA receptor encephalitis is associated with cerebrospinal fluid (CSF) IgG antibodies against the GluN1 subunit of the NMDA receptor. Since antibody tests are not accessible in many institutions and it can take several weeks to obtain the results, most recent studies have aimed to improve knowledge of psychopathology and the clinical specificities of these diseases [25,26,27,28,29,30]. The clinical and paraclinical criteria have been defined to identify a probable anti-NMDA receptor encephalitis as soon as possible [31], but may not be sufficient in cases of autoimmune encephalitis without neurologic symptoms, and a systematic serum anti-NMDA-r antibodies test, electroencephalogram (EEG), and magnetic resonance imaging (MRI) of the brain are recommended [32]. For patients with psychotic symptoms presenting with clinical criteria or MRI or EEG signs, routine CSF should be mandatory [33,34].

#### 3.2.2. Systemic Lupus Erythematosus

Systemic Lupus Erythematosus (SLE) is a prototype of a chronic, inflammatory, systemic autoimmune disease with unknown etiology, preferentially affecting females in their childbearing years [35,36], but two-thirds of pediatric patients develop neuropsychiatric symptoms [37]. Neuropsychiatric SLE (NPSLE) includes the neurologic syndromes of the central, peripheral, and autonomic nervous systems and psychiatric symptoms including depression, psychosis, and anxiety [37,38,39]. Delusions and hallucinations are the two main symptoms of SLE psychosis; however, they can also occur after administration of corticosteroids, and a differential diagnosis between NPSLE and cortico-induced neuropsychiatric disorders must further be considered [40]. The diagnosis of SLE is based on clinical and paraclinical criteria, but the lack of pathognomonic symptoms or test findings makes the diagnosis of NPSLE difficult to confirm. According to the last classification criteria for SLE, positive antinuclear antibodies are an obligatory criterion [41].

#### 3.2.3. Hashimoto Encephalopathy

Two articles concerned Hashimoto encephalopathy (HE), a rare neuropsychiatric syndrome associated with Hashimoto thyroiditis, an autoimmune-mediated chronic inflammation of the thyroid. HE is more common in women and is associated with serologic evidence of antithyroid antibodies when other causes of encephalopathy are excluded [42]. In these cases, clinical manifestations include neurologic symptoms such as seizures or abnormal movements, impaired consciousness, and behavioral changes. Additionally, psychiatric symptoms including aggression, severe irritability, hallucinations, insomnia, and agitation have been described as dominant clinical features for all pediatric patients examined [43]. Brain MRI and EEG are recommended in all patients to account for the differential diagnosis of other possible encephalopathies, but the results are not specific to HE. The diagnosis is based on the identification of clinical manifestations that are associated with increased antithyroid antibodies, and all suspected patients presenting with acute or subacute encephalopathy or cognitive decline of unknown etiology should be tested for antithyroid antibodies in serum or CSF [44]. When the antibody titer is normal at presentation with persistent symptoms of HE and evidence for another diagnosis, follow-up determinations of antithyroid antibody titers can lead to a confirmed diagnosis [43].

#### 3.2.4. Acute Disseminated Encephalomyelitis

Acute disseminated encephalomyelitis (ADEM) is an immune-mediated inflammatory process involving central nervous system white matter that mainly occurs in children [45]. ADEM typically follows a viral infection, but it can occasionally occur without any defined preceding trigger. The classical presentation is characterized by the acute onset of neurologic abnormalities. Several cases of initial presentation with acute psychosis have been reported. In these cases, fever and focal neurologic signs were absent on presentation [46]. Early MRI facilitates the diagnosis [47] and often reveals diffuse, symmetric white matter demyelinating lesions [45,48].

### 3.3. Metabolic Diseases

Inborn errors of metabolism (IEMs) are physiological defects or malfunctions due to full or partial loss of gene function, usually caused by autosomal recessively inherited enzyme defects [13]. Although metabolic diseases are not frequent, the number of articles concerning these pathologies has greatly increased over the last ten years. Their impact on the central nervous system can be responsible for psychiatric syndromes such as psychosis, depression, anxiety, or mania [49,50]. Niemann-Pick disease and Wilson’s disease were the two main metabolic disorders described in the literature.

#### 3.3.1. Niemann–Pick Disease

Niemann–Pick disease type C (NP–C) is a rare (<1/100,000 births) [51], autosomal recessive neurodegenerative disease caused by mutations in the NPC1 or NPC2 gene that lead to impaired intracellular lipid trafficking and excess storage of cholesterol and glycophospholids in the brain, liver, and other tissues [52]. Patients with NP–C show extreme clinical heterogeneity in their symptom profiles, ages at disease onset, and rates of disease progression [53]. Neurological disease onset typically occurs during childhood [53], and psychiatric manifestations are frequently reported with a high incidence of psychotic symptoms, comprising paranoid delusions, auditory hallucinations, and disorganized thoughts [54,55]. Psychiatric manifestations may remain isolated for several years, and most patients who initially present with psychotic symptoms do not have obvious abnormalities on neurological examination [56,57]. Although several biochemical markers have been evaluated to provide a low-invasive and inexpensive diagnostic method [58], the definite diagnosis of NP–C disease remains based on the demonstration of specific mutations in the genes NPC1 and NPC2 [59].

#### 3.3.2. Wilson’s Disease

Wilson’s disease is an autosomal recessive disease that is more common than NP–C disease (6/100,000) [57] and commonly affects children or young adults [59,60]. It is caused by a mutation of a gene [ATP7B] encoding for a copper transportation protein, leading to copper accumulation in the liver, kidney, bones, and brain. Psychiatric symptoms appear early, and it has been estimated that one in two patients present psychiatric signs in the absence of other organic signs [61]. The most frequently reported psychiatric manifestations are mood disorders, including both depressive and manic elements [62,63,64,65,66], hallucinations [67,68,69], and personality changes with the emergence of irritability and aggressive behavior [70,71]. A meta-analysis found a 2.4% frequency of “frank psychosis” resembling schizophrenia [60]. Blood copper measurement and the existence of a conventional Kayser–Fleisher ophthalmic ring are important diagnostic indicators of Wilson’s disease. The current diagnosis is made through MRI findings indicating thalamic and lenticular nucleus hyper-signals [57].

#### 3.3.3. Disorders of Homocysteine Metabolism

Homocysteine is an essential amino acid and is metabolized along pathways of remethylation or transsulfuration. Disorders of homocysteine metabolism (DHMs) are usually caused by the absence of an enzyme in one of these pathways. Two distinct enzymes are concerned: (1) methyltetrahydrofolate reductase deficiency in the remethylation pathway and (2) cystathionine beta synthase (CbS) deficiency in the transsulfuration pathway. These deficiencies lead to a functional deficiency of folate or B12 despite normal circulating levels [57,59]. DHMs are responsible for highly variable clinical features such as skeletal abnormalities, neurologic signs, and psychiatric problems that can essentially be characterized as psychotic symptoms [57,72]. Psychotic-like symptoms can be an isolated clinical feature [73] and often produce visual hallucinations [74]. Brain imaging sometimes shows demyelination but may also present as completely normal. An MRI scan of the spinal cord may show high signal intensity in the dorsal columns of the spinal cord, similar to that seen in pernicious anemia [59]. The diagnosis is based on amino acid chromatography and the determination of homocysteine [57].

#### 3.3.4. Acute Intermittent Porphyria

Acute intermittent porphyria (AIP) is an autosomal dominant disease with variable penetrance that is linked to the deficiency of porphobilinogen deaminase, an enzyme that is involved in the biosynthesis of heme [59]. Clinical signs of AIP usually appear in adults, but cases of childhood onset have been reported [57,75,76]. The “classical triad” of symptoms consists of abdominal pains, psychiatric disturbances, and peripheral neuropathies. Psychiatric symptoms such as delusions, hallucinations, and the development of thought disorders [57] are frequent [75,77] and can be the single presenting feature [49,78]. Acute attacks are often triggered by porphyrinogenic treatments (contraceptives, barbiturates, sulfonamides, antiepileptics), sepsis, or alcohol ingestion [76]. The diagnosis of AIP should be considered in any psychiatric syndrome with unexplained cyclical pain [79]. This diagnosis is based on the measurement of delta-aminolevulinic acid in blood testing and porphibilinogene in urinalysis [57], which may be normal between attacks [59].

#### 3.3.5. Cerebrotendinous Xanthomasis

Cerebrotendinous xanthomatosis (CTX) is a hereditary metabolic disorder characterized biochemically by sterol 27-hydroxylase deficiency, an enzyme involved in the degradation of cholesterol. This metabolic deficiency causes the formation of xanthomatous lesions (gradual buildup of cholestanol, a cholesterol metabolite) in various tissues, including the brain and tendons [59]. The estimated prevalence of CTX is <5 in 100,000, and it varies by country and ethnic group [80,81]. Clinically, patients usually have juvenile cataracts, tendinous xanthomata, and xanthomas (lipid accumulation in the superficial dermis) as visible signs [57]. Neurological signs and psychiatric disorders (psychotic manifestations, hallucinations) are often associated with CTX [82] and appear during adolescence. Acute psychotic-like episodes have been described, even though other behavioral disorders, especially attention-deficit hyperactivity disorders, are more frequent in childhood and adolescence [82]. In CTX cases, MRI shows a typical dendritic hyper-signal in the cerebellar nuclei regions. The diagnosis is confirmed by plasma cholestanol measurements and gene sequencing [57].

#### 3.3.6. Arylsulfatase A Deficiency

Arylsulfatase A Deficiency, also known as metachromatic leukodystrophy (MLD), is an autosomal recessive disorder in which the partial or complete absence of arylsulfatase A leads to demyelination of the central and peripheral nervous systems [83]. The clinical presentation is heterogeneous with respect to the age of onset, the rate of progression, and the initial symptoms [84,85,86]. The neurological signs of MLD are often preceded by precursor symptoms in childhood, mainly behavioral disturbances and psychotic symptoms such as auditory hallucinations, bizarre delusions, and catatonic posturing [87]. The diagnosis should be suspected in individuals with progressive neurologic dysfunction, MRI evidence of leukodystrophy, or low arylsulfatase A activity in leukocytes [84]. The presence of sulfatides in urine is a useful indicator of the enzyme block [56]. The diagnosis of MLD can also be confirmed by molecular testing or after the identification of metachromatic lipid deposits in a nerve or brain biopsy [84].

#### 3.3.7. Hereditary Transthyretin Amyloidosis

Hereditary transthyretin amyloidosis (ATTR) is a systemic disorder characterized by the extracellular deposition of misfolded transthyretin protein [88]. The clinical presentation is typically a slowly progressive sensorimotor and/or autonomic neuropathy [89] that is frequently accompanied by non-neuropathic features such as cardiomyopathy, nephropathy, vitreous opacities, and central nervous system (CNS) amyloidosis [90]. Individuals with leptomeningeal amyloidosis (LA) show CNS signs, including psychotic symptoms. The diagnosis of ATTR is established with the above clinical features, a biopsy showing amyloid deposits that bind to anti-TTR antibodies, and the identification of a heterozygous pathogenic variant in TTR by molecular testing [90]. In LA, the protein concentration in the CSF is usually high, and gadolinium-enhanced MRI typically shows an extensive increase in the surface of the brain, ventricles, and spinal cord [91].

#### 3.3.8. Urea Cycle Disorders

Urea cycle disorders are caused by inherited enzyme deficiencies. Ornithine transcarbamylase enzyme deficiency is the most common of these diseases (1 in 8000 births) [92]. These disorders may evolve chronically during the neonatal period with mild symptoms until a first acute hyperammonemia episode appears. Patients present with symptoms related to the accumulation of ammonium in the body, including digestive, neurological, and psychotic symptoms [93]. These pathologies can easily be revealed by a measurement of plasma ammonemia.

#### 3.3.9. Tay-Sachs Disease

Tay-Sachs disease (TSD) is an autosomal recessive lipid storage disorder caused by ß-hexosaminidase A deficiency. The prevalence of TSD is estimated to be 1 in 200,000 live births [94]. Three clinical variants of TSD, based on the age of onset, have been described in the literature [56]. Psychiatric signs may be the only manifestations of the chronic form, which usually presents in late childhood or adolescence; acute psychosis is reported in 30% to 50% of cases [95,96,97]. The presence of speech disturbances, gait abnormalities, movement disorders, and cognitive decline may indicate an underlying metabolic disorder. Most reports suggest that neuroleptic medications are rarely efficacious and may produce an unacceptably high risk/benefit ratio, whereas benzodiazepine may ameliorate the psychiatric and neurologic abnormalities in these patients [56,95,96].

### 3.4. Neurologic Diseases

#### 3.4.1. Huntington’s Disease

Huntington’s disease (HD) is an incurable, autosomal dominant, neurodegenerative disease caused by an expanded CAG repeat in the huntingtin gene [98]. In HD, the production of a mutant protein leads to progressive motor impairment, psychiatric disorders, and cognitive decline [98,99]. The first manifestations begin around midlife. Neuropsychiatric symptoms, including psychosis, are highly prevalent and appear in the premotor phase [100]. According to the current criteria, the diagnosis is centered around the presence of involuntary choreiform movements and a positive genetic test for the CAG-expanded allele gene or a family history of HD [101].

#### 3.4.2. Frontotemporal Dementia

Frontotemporal dementia (FTD) is a neurodegenerative disorder indicating a focal clinical syndrome; it is characterized by changes in one’s personality and social conduct with circumscribed degeneration of the prefrontal and anterior temporal lobes [102]. Clinical and clinicopathological observations have found that people with younger-onset FTD frequently present with symptoms of schizophrenia, bipolar disorder, and affective disorders. When patients who are diagnosed with schizophrenia have an insidious and evolving cognitive deficiency, the probability of FTD is higher [103]. The characterization of FTD remains challenging, and, in the absence of definitive biomarkers, the diagnosis is based on clinical criteria including early behavioral disinhibition, apathy, perseverative behaviors, hyperorality, and executive deficits [104,105,106].

#### 3.4.3. Wernicke Encephalopathy and Korsakoff’s Psychosis

Wernicke encephalopathy (WE) is an acute neuropsychiatric condition caused by thiamine (vitamin B1) intracellular depletion in brain cells [107]. Alcohol misuse is the first cause of thiamine deficiency in industrialized countries but can also be seen in depleted teenage mothers in areas of poverty [108,109]. Korsakoff’s psychosis results from inadequate treatment or a failure to diagnose WE in a timely manner and is characterized by severe short-term memory loss and hallucinations [110]. MRI investigations show thalamic, mamillary body, and frontal lobe atrophy, but the diagnosis remains based on clinical features and can be supported by a history of alcohol abuse [111,112].

#### 3.4.4. Kleine–Levin Syndrome

Kleine–Levin syndrome (KLS) is a rare disease with an estimated prevalence of 1–2 per million people in France [113,114]. The triad of hypersomnia, megaphagia, and hypersexuality is classically reported in KLS cases, but the clinical presentation is rather hypersomnia with prolonged sleep times accompanied by a combination of cognitive, behavioral, and/or psychiatric disturbances. These symptoms also include depression, anxiety, and, rarely, suicidality [115]. Derealization, dissociation (feelings of mind-body disconnection), and altered perceptions are frequently described by patients with KLS [116]. The symptomatic periods alternate with intermitting periods of normalcy. Sleep monitoring and functional brain imaging during and between episodes are useful to support the diagnosis, which is based on clinical features [117,118].

#### 3.4.5. Narcolepsy

Narcolepsy is a chronic sleep disorder that affects the regulation of sleep-wake cycles. It occurs in approximately 1 in 2000 individuals [119] and usually begins during adolescence. The classical symptoms of narcolepsy are excessive daytime sleepiness and sleep attacks, and the diagnosis is based on physiological testing, including nocturnal polysomnography and daytime EEG measurement of sleep latency. Hypnagogic/hypnopompic hallucinations are also reported, and the distinction between this presentation and the hallucinations of schizophrenia may be challenging in some clinical settings [119,120]. Most studies suggest that these psychotic symptoms are worsened by antipsychotic drugs; by contrast, they abate when narcolepsy is identified and treated with psychostimulants [119].

### 3.5. Infectious Diseases

A viral hypothesis for schizophrenia was seriously considered at the beginning of the 20th century [121]. More recent data show that some infectious agents, like toxoplasma gondii or CMV, could explain psychosis pathophysiology; however, a clear mechanism and a chronology are still difficult to establish. However, the human immunodeficiency virus appears to take a central role in these infectious etiologies [122], which are caused by well-known opportunistic agents [123].

#### 3.5.1. Human Immunodeficiency Virus

Human Immunodeficiency Virus (HIV) is estimated to be associated with psychotic symptoms in 0.23% to 15% of cases [14]. These symptoms, characterized by persecutory, grandiose, and somatic delusions with hallucinations, present generally either in late-stage HIV or when patients have transitioned to acquired immunodeficiency syndrome (AIDS) [124,125]. One complicating factor in assessing this association is the fact that side effects of antiretrovirals include hallucinations [125]. HIV is diagnosed by the detection of antibodies (anti-HIV) in serum and confirmed by a Western blot test. In addition to a specific link between HIV and psychosis, some opportunistic infections related to HIV also trigger psychotic manifestations. Moreover, patients with HIV are at increased risk of developing extrapyramidal symptoms and tardive dyskinesia, particularly with the use of atypical antipsychotics [124,126]. Systematic testing (detection of anti-HIV antibodies in serum) is recommended in the case of FEP, but consent from each patient may be required, according to some state laws, before testing [127].

#### 3.5.2. Toxoplasmosis

Toxoplasma gondii is an intracellular protozoan whose definitive host is the cat. Humans are infested by ingestion of contaminated food or water, or by congenital transmission to the fetus [128]. In immunocompetent hosts, primary infection typically remains asymptomatic. Toxoplasma is the most common infection in HIV-seropositive immunocompromised hosts, and in some cases, patients present with neurological or psychotic symptoms [129,130]. An MRI may show specific cerebral granuloma lesions, and the diagnosis is confirmed by serologic testing [129,131].

#### 3.5.3. Cytomegalovirus

Cytomegalovirus (CMV) is a beta-herpesvirus that spreads by personal contact or congenital transmission. Most initial CMV infections are asymptomatic in immune-competent individuals, but reactivation may occur in cases of immunodeficiency (transplantation, HIV infection) and lead to psychiatric symptoms [130]. Clinical symptoms include deficits in concentration, memory, manipulation of knowledge, humor, and emotional expression [132]. A case of auditory hallucinations, delusions, and tangential thinking has also been described in the literature [133]. The diagnosis of CMV is confirmed by serologic testing.

#### 3.5.4. Syphilis

Syphilis is a chronic, sexually transmitted disease caused by Treponema pallidum [134]. In the absence of adequate treatment, the disease follows several stages: primary, secondary, and tertiary syphilis. Neurosyphilis can occur at any stage of the disease but is classically associated with tertiary syphilis [134]. The clinical manifestations of neurosyphilis are not specific, and patients may present with psychotic symptoms mimicking psychiatric illness [14,135]. The diagnosis can be suspected based on clinical findings. The demonstration of T. pallidum in infected tissues or fluid confirms the diagnosis, but serologic testing remains the mainstay of laboratory diagnosis [135,136].

#### 3.5.5. Lyme Borreliosis

Lyme borreliosis is a bacterial disease caused by a spirochete, Borrelia burgdoferi, which is transmitted by ticks. Its prevalence is correlated with the presence of forest areas [137]. If not treated at the initial stage, characterized by the apparition of erythema migrans, spirochetes spread in the organism and infect various organs, especially the nervous system. Neurological symptoms can occur several years after the initial infection, as spirochetes can remain quiescent for a long time [138]. The symptoms of Lyme neuroborreliosis (LNB) include encephalopathies and encephalomyelitis, but mood disturbances and psychotic symptoms are also described [139,140]. The CSF analysis is a key test for the diagnosis of LNB, which is based on several criteria defined in the article by Mygland et al. [141].

### 3.6. Genetic Diseases

The relationship between some genetic disorders and psychotic-like symptoms has been noted since the 1980s [142]. Because of the multiplicity of possible gene alterations and the low prevalence of these diseases, however, systematic genetic screening is not yet conceivable. However, genetic screening should be discussed in the case of a clinical presentation evocative of a genetic disorder. In most of these diseases, patients suffer from several physical disorders and may present with an intellectual disability, dysmorphia, or resistance to antipsychotic treatments. The genetic syndromes reported in the literature are 22q11 deletion syndrome, Prader–Willi syndrome, C9orf72 mutations, Fragile X premutation disorders [143,144], BSC1L mutation [145], 3q29 recurrent deletion [146], 15q duplication syndrome [147], and genetic mitochondrial disorders [148]. The syndromes that are most frequently found in cases of FEP are described below.

#### 3.6.1. 22q11 Deletion Syndrome

22q11 deletion syndrome is a common (1 in 4000 births) genetic disorder. Patients suffer from several physical disorders, such as congenital heart defects, palate defects, hypoparathyroidism, and immunologic abnormalities. This syndrome is also associated with psychiatric manifestations [149,150], and 23–32% of adults present with psychotic disorders [151,152]. This deletion is estimated to account for 1–2% of schizophrenia cases overall [153].

#### 3.6.2. Prader–Willi Syndrome

Prader–Willi syndrome (PWS) is a neurodevelopmental disorder resulting from a genetic anomaly on chromosome 15, with two main subtypes identified. The birth incidence is estimated at around 1 in 22,000 [154]. Patients with PWS usually present with an association of clinical features such as hypotonia, hypogonadism, hyperphagia, mild learning disabilities, and small hands and feet. A comorbid association of PWS with psychiatric disorders has been suggested in several studies [155,156]. In both genetic subtypes, psychiatric manifestations are mainly atypical affective disorders with or without psychotic symptoms [157,158].

#### 3.6.3. C9orf72 Mutations

C9orf72 mutations have been found in almost 12% of patients with FTD and are also associated with some forms of amyotrophic lateral sclerosis (ALS) [159]. In patients with FTD or ALS, late-onset psychotic disorders as the initial presentation have been reported to be more frequent when these diseases are secondary to the C9orf72 mutation [160,161,162].

## 4. Discussion

This systematic review reveals that many somatic illnesses, most of which may occur or begin during childhood or adolescence, can be responsible for psychotic symptoms. Determining the appropriate initial work-up implies that the prevalence of these pathologies and screening test features are to be taken into account [127].

### 4.1. Limitations

Studies published for the past fifteen years were selected to illustrate less-investigated, actual, and contemporary concerns about FEP. Well-known differential diagnoses (i.e., blood sugar troubles) do not require active academic research. As a consequence, these pathologies did not appear in this systematic review as they are systematically screened in current practice. As this literature review focused on clinical perspectives, the bibliographic search was restricted to medical databases. However, the probability that our pathology list will be representative of current research concerns is increased by the inclusion of the only review that has been published to date.

### 4.2. Prevalence

Diseases in which psychosis is a typical presentation (i.e., Wilson’s disease, thyroid diseases, etc.) should be systematically screened for FEP [127]. Many of the diseases found in this review can potentially influence psychotic symptoms even if they are not the usual presentation, especially as some of these pathologies are very rare. It would be arduous, if not infeasible, to systematically screen every disease in which psychotic symptoms have been reported; however, it seems necessary to propose an initial work-up that will help physicians who suspect a somatic origin of a given disorder.

### 4.3. Causality

When a somatic disease is found in a patient with FEP, the question is whether this pathology is the cause of psychosis or a comorbidity. The stress-diathesis model of schizophrenia shows this distinction is difficult to make: a somatic disorder may act as an external stress factor in individuals with pre-existing vulnerability factors [163,164]. However, models have been proposed to assess whether an association can causally contribute to a pathological phenomenon [165,166], and three key principles (atypicality, temporality, and explicability) have been suggested to help establish a cause-and-effect relationship between a somatic condition and psychotic symptoms [14]. In an atypical presentation of FEP, psychotic symptoms that appear after the onset of a medical condition and the possibility of establishing a physiological link between a somatic lesion and psychotic symptoms may provide evidence of a secondary psychotic disorder.

### 4.4. Exams

#### 4.4.1. Laboratory Tests

A consensus exists on the performance of laboratory screening, but opinions differ on how broad this screening should be [1,127,167,168]. Many diseases reported in this review may produce inflammation that can be detected with a C-reactive protein test.

An ammonemia measurement, which is a low-cost exam, should be systematically performed to eliminate hyperammonemia. Positive antinuclear and anti-NMDA antibodies associated with an inflammatory syndrome may reveal an autoimmune disorder such as SLE or limbic encephalitis. A negative syphilis serology allows for ruling out neurosyphilis. This serologic test may also show an underlying autoimmune disease in the case of false-positive results [169].

A TSH test is necessary to reveal thyroid dysfunction. It may also lend support in detecting HE [170]. Suspicion of Wilson’s disease, which is the most frequent metabolic disorder responsible for psychotic symptoms, can be formally excluded with a negative result of the copper measurement in the blood.

All infectious diseases outlined in this review, except syphilis and LNB, appeared in cases of immunodeficiency in HIV-seropositive hosts. As the HIV-seropositive condition can itself be associated with psychotic manifestations, systematic detection is recommended in cases of FEP after recording consent from patients [14,129,130,171].

Additionally, this review highlights the role of genetic diseases, particularly the relationship between psychotic disorders and 22q11 deletion syndrome. Clinicians should systematically seek this genetic mutation when patients present with clinical manifestations such as a developmental delay or facial dysmorphology [149,172].

The main function of an electrocardiogram (ECG) is to identify a cardiac disease, which is of particular importance before introducing an antipsychotic treatment [1]. Conduction disorders detected in young patients should also lead clinicians to suspect lupus [173].

EEG is an inexpensive, easily available, and noninvasive exam that is helpful for the diagnosis of limbic encephalitis when specific signs are detected (extreme delta brush waves have been reported in 30% of cases) [16,31,174,175].

Most of the diseases reported in this review can be revealed by magnetic resonance imaging (MRI): signs of encephalitis [43,44], leukodystrophy [56], white matter lesions [173], diffuse cortical atrophy [162], or specific lesions of toxoplasmosis [129]. Even though cerebral computed tomography (CT) is considerably more feasible, this exam uses ionizing radiation. CT is also less sensitive than MRI in the diagnosis of cerebral lesions [3]. Moreover, studies report good patient compliance, and an economic analysis suggested that MRI could be cost-saving when the prevalence of organic causes is around 1%, while the prevalence of imaging findings that would influence clinical management is estimated at around 5% in the case of FEP [3,10,168,170,176].

#### 4.4.2. Lumbar Puncture

This review emphasizes the increasing number of limbic encephalitis cases reported and the high probability of underdiagnosis of this pathology [10,177,178]. Considering these observations, lumbar puncture would be indicated in the FEP initial work-up [179,180]. The invasive aspect of this exam is highly counterbalanced by the opportunity to diagnose a curable disease and consider the risk of a paradoxical reaction to antipsychotics in the case of encephalopathy.

## 5. Conclusions

This review highlights that the timely diagnosis of a physical disease is fundamental when a patient presents with psychotic symptoms in order to (1) administer the appropriate therapy and (2) avoid paradoxical reactions to antipsychotics. This differential diagnosis is possible only if the appropriate exams are performed. The initial exams recommended are presented in Table 1. Specifically, our results indicate that laboratory tests, including the antinuclear antibody test, the test of anti-NMDA antibodies, and HIV testing, must be performed systematically. The benefit-risk ratio indicates that brain MRI should be considered in cases of FEP. Finally, lumbar puncture (which is an invasive exam) could be of interest for some patients, considering the clinical presentation. Future studies should provide additional data on the feasibility and acceptability of these exams, particularly lumbar punctures.

## Figures and Tables

**Figure 1 children-10-01439-f001:**
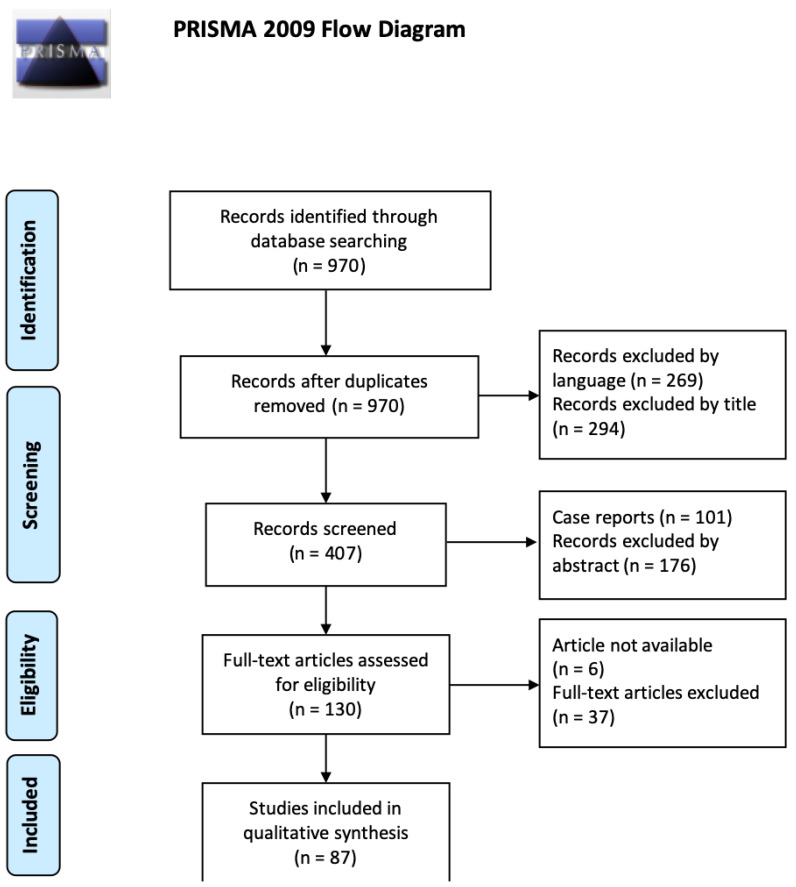
Prisma flow diagram for the literature search.

**Table 1 children-10-01439-t001:** Initial exams recommended in the case of a first episode of psychosis.

Laboratory tests	C-reactive protein test TSH Ammonemia Ceruloplasmin FTA-ABS (fluorescent treponemal antibody absorbed) HIV test (after obtaining consent from patients) Antinuclear and anti-NMDA antibodies
Imaging	Brain MRI
Genetic tests	Karyotype and research of specific mutations, such as a 22q11 deletion, for patients presenting physical or developmental features associated with psychiatric manifestations
Others	ECG EEG Lumbar puncture in search of pleocytosis or oligoclonal bands in the CSF

## Data Availability

The data presented in this study are available in Appendix A.

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
