# Peer review of "Psychosis Caused by a Somatic Condition: How to Make the Diagnosis? A Systematic Literature Review"

_children, 2023, doi:10.3390/children10091439_

Round 1

Reviewer 1 Report

This manuscript entitled "PSYCHOSIS CAUSED BY A SOMATIC CONDITION: HOW TO MAKE THE DIAGNOSIS? A SYSTEMATIC LITERATURE REVIEW" deals with the clinically important topic of differentiating between endogenous psychosis and psychotic symptoms caused by physical illness. While this manuscript is of high interest to many, especially clinicians, this manuscript seems to include the following concerns.  

  1. 1) In the introduction, the authors described the association between psychotic-like symptoms and physical illness as well as FEP. The significance of this study can be enhanced by mentioning previous research on the association of physical illness in individuals with ultra-high risk and psychotic-like experiences.  

  1. 2) In the introduction, it would be better to describe how the exclusion of physical disease is addressed in the current published guidelines for dealing with FEP.  

  1. 3) Clarification on the definition of FEP in this study would be desirable. Psychotic symptoms and FEP are two different concepts.  

  1. 4) It seems inconsistent that Hashimoto encephalopathy is listed in the results, even though it was stated in the method that thyroid disease was not included.  

  1. 5) In the results, a table regarding the list of diseases would be helpful to understand.  

  1. 6) The authors state that "laboratory tests including the antinuclear antibody test, the test of anti-NMDA antibodies, and HIV testing must be performed systematically," however, this seems to be an excessive interpretation of the results of this study. What do the authors mean by "systematically"? Furthermore, while it is obviously important not to miss these diseases, not all medical facilities can conduct these tests routinely, and suggestions that are in line with the actual situation are needed.

Author Response

We thank the reviewers for accepting to consider the proposal to resubmit our article entitled “Psychosis caused by a somatic condition: how to make the diagnosis? A systematic literature review”.

You will find below the responses to each comment. The revised manuscript has been uploaded with revision marks.

We hope the revised manuscript will meet your expectations for publication in Children.

Point by point response to reviewer 1:

This manuscript entitled "PSYCHOSIS CAUSED BY A SOMATIC CONDITION: HOW TO MAKE THE DIAGNOSIS? A SYSTEMATIC LITERATURE REVIEW" deals with the clinically important topic of differentiating between endogenous psychosis and psychotic symptoms caused by physical illness. While this manuscript is of high interest to many, especially clinicians, this manuscript seems to include the following concerns.  

>> 1) In the introduction, the authors described the association between psychotic-like symptoms and physical illness as well as FEP. The significance of this study can be enhanced by mentioning previous research on the association of physical illness in individuals with ultra-high risk and psychotic-like experiences.  

We thank the reviewer for this remark. Previous research established the association between physical illness and psychotic-like experiences in subclinical populations (Sleurs, D.; Dubertret, C.; Pignon, B.; Tebeka, S.; Le Strat, Y. Psychotic-like Experiences Are Associated with Physical Disorders in General Population: A Cross-Sectional Study from the NESARC II. Journal of Psychosomatic Research 2023, 165, 111128). We mentioned this association as follows: “Furthermore, several studies have found that psychotic-like experiences may be associated with physical disorders in ultra-high-risk patients [4] and that at least 5% of FEP could be caused by a physical disease [5,6].”

>> 2) In the introduction, it would be better to describe how the exclusion of physical disease is addressed in the current published guidelines for dealing with FEP.  

                We thank the reviewer for this remark. Guidelines for FEP recommend a “general medical assessment” to exclude a physical disease but they do not specify which pathologies can lead to psychotic symptoms and there is no consensus for a comprehensive medical assessment. We addressed this point as follows:  “However, the low specificity of FEP symptoms makes detection highly challenging, and, even though guidelines propose a “general medical assessment” [1] no consensus of recommendation has been established for a comprehensive medical assessment to eliminate somatic causes.”

>> 3) Clarification on the definition of FEP in this study would be desirable. Psychotic symptoms and FEP are two different concepts.

                We thank the reviewer for this remark. According to the Australian guidelines for early psychosis, a first episode of psychotic disorder is defined as a “full threshold disorder with moderate–severe symptoms, neurocognitive deficits and functional decline (GAF 30–50)” and “includes acute and early recovery periods”. We clarified the definition of FEP as follows: “First episode of psychosis (FEP) is a full threshold clinical condition in which altered perceptions of reality, delusions, disorganized thoughts, and cognitive dysfunctions can be observed by others and are associated with functional decline [1].”

>> 4) It seems inconsistent that Hashimoto encephalopathy is listed in the results, even though it was stated in the method that thyroid disease was not included. 

We thank the reviewer for this remark. Although psychotic manifestations of common thyroid disorders are well known, Hashimoto encephalopathy, which is associated with Hashimoto thyroiditis, is a rare syndrome and specific tests are required to make the diagnosis. In the eligibility criteria, we specified “common thyroid disorders”: “Psychotic presentation of medical conditions such as brain tumors, traumatic brain disorders, substance abuse, and common thyroid disorders has been clearly established for several decades [1,13,14].”

>> 5) In the results, a table regarding the list of diseases would be helpful to understand.

                We thank the reviewer for this remark. We have added a table listing the main symptoms and paraclinical exams recommended for each pathology (Appendix B).

>> 6) The authors state that "laboratory tests including the antinuclear antibody test, the test of anti-NMDA antibodies, and HIV testing must be performed systematically," however, this seems to be an excessive interpretation of the results of this study. What do the authors mean by "systematically"? Furthermore, while it is obviously important not to miss these diseases, not all medical facilities can conduct these tests routinely, and suggestions that are in line with the actual situation are needed.

We thank the reviewer for this remark. The diagnosis of limbic encephalitis can be difficult, as psychotic as psychotic symptoms may be the only ones presented by the patient, in which case anti-NMDA antibody testing is the only way to make the diagnosis. In addition, the risk of infection is too often overlooked, particularly in the case of HIV infection. The fact that at least 5% of FEP is caused by a physical disease justifies the systematic measurement of antibodies and HIV testing. When these tests are not available, patients should be referred to another medical center.

Reviewer 2 Report

Dissaux et al. provide a systematic review of various disorders that may offer a differential somatic diagnosis for the first episode of psychosis and provide general recommendations for how this differential diagnosis may be achieved. Overall, I was impressed with the scope of the review and feel the final list of recommendations provided by the authors is of import to the field. Please note that I received access to a “v2” of the manuscript, to me this seems to indicate the authors have provided an updated version of the manuscript after an initial round of peer review, though as I did not have access to any previous reviewer comments or responses, I am also not entirely sure of this. Under the assumption this manuscript has already gone through one round of peer review, I’ve tried to limit my comments to the most vital ones:

Search strategy & methodology: The search strategy describes that “review” was a search criterion and many of the commonalities between studies, such as the number of studies, etc.) also suggest that the authors were only interested in identifying review articles and not original publications/research articles. However, that the systematic review as performed by the authors is in fact a review of reviews (or umbrella review) is not made apparent in the abstract or the final paragraph of the introduction. However, from the discussion (page 12 lines 499-502) I understood that this manuscript was not an umbrella review but had as a focus clinical perspectives and only included a single recent review. Moreover, on page 3 lines 89-90, it states the authors will exclude case reports; however, Appendix A (pages 14-34) describes some of the included manuscripts as cast reports and contains numerous references to various types of review studies. Putting all the information about the search criteria and methodology together left me quite confused about what the authors aimed to do and had achieved. All sections concerning the methodology need a rigorous rewrite to clarify what was actually done. Lastly, as a minor note, I think page 3 line 121 should contain a reference to figure 1, but now contains an error message, and the actual figure has “table 1” as a legend, which I don’t feel is appropriate.

Readability and understandability for a broad audience: First, the description of the results for limbic encephalitis immediately starts with an overview of associated antibodies but does not entail a brief description of what it is (page 4, lines 128-129). This contrasts with other sections where some form of brief description is given. As I believe the readership of this journal may well be highly diverse, I would generally recommend giving more general descriptions of what the various disorders detail as the level of detail provided at present differs from disease to disease. Second, from the sentence on page 10 lines 407-409, it is unclear what kind of systematic testing is required. Moreover, I assume that consent for testing is a more general criterion, not limited to this specific setting, while the reference to state laws suggests this may be US-specific. All in all, this sentence is hard to understand without returning to the cited article. Similarly, on page 11 lines 447-448 “CSF analysis is a key yes for the diagnosis of LNB, ich is defined as follows [ref 140]” cannot be understood without reading the cited articles. Overall, I recommend the authors to carefully check the manuscript for similar sentences and provide adequate information to understand the point they are trying to make without the need to read the cited articles. Third, in section 3.6 genetic diseases (page 11, lines 450-485), unlike the previous sections, the recommendations for differential diagnosis are lacking. While even lay people will understand this will require some form of genetic screening, for consistency across the review as well as for clarity, I recommend the authors to either add a general statement on how genetic screening may be achieved to the introductory paragraph of this section or to describe the specific, preferred genetic test for each of the disorders. Finally, table 2 (page 13 line 575) summarizes the types of testing the authors recommend in section 4.4 exams (pages 12-14, lines 528-576), but no reference to this table is made in the main text.

While overall the paper is written in clear English, I’ve nonetheless ticked the “minor editing of English required”, e.g., throughout the manuscript the Oxford comma is not always applied correctly, for some terms the full term plus abbreviation is given multiple times in the manuscript (sometimes in quick succession, see e.g., page 9-10 lines 392-393 and line 397), for other terms abbreviations are given but never used after their introduction, while yet other terms are never given in full. Final checks of the manuscript for proper use of the Oxford comma and abbreviations are recommended. 

Author Response

We thank the reviewers for accepting to consider the proposal to resubmit our article entitled “Psychosis caused by a somatic condition: how to make the diagnosis? A systematic literature review”.

You will find below the responses to each comment. The revised manuscript has been uploaded with revision marks.

We hope the revised manuscript will meet your expectations for publication in Children.

Point by point response to reviewer 2:

Dissaux et al. provide a systematic review of various disorders that may offer a differential somatic diagnosis for the first episode of psychosis and provide general recommendations for how this differential diagnosis may be achieved. Overall, I was impressed with the scope of the review and feel the final list of recommendations provided by the authors is of import to the field. Please note that I received access to a “v2” of the manuscript, to me this seems to indicate the authors have provided an updated version of the manuscript after an initial round of peer review, though as I did not have access to any previous reviewer comments or responses, I am also not entirely sure of this. Under the assumption this manuscript has already gone through one round of peer review, I’ve tried to limit my comments to the most vital ones:

>> Search strategy & methodology: The search strategy describes that “review” was a search criterion and many of the commonalities between studies, such as the number of studies, etc.) also suggest that the authors were only interested in identifying review articles and not original publications/research articles. However, that the systematic review as performed by the authors is in fact a review of reviews (or umbrella review) is not made apparent in the abstract or the final paragraph of the introduction. However, from the discussion (page 12 lines 499-502) I understood that this manuscript was not an umbrella review but had as a focus clinical perspectives and only included a single recent review. Moreover, on page 3 lines 89-90, it states the authors will exclude case reports; however, Appendix A (pages 14-34) describes some of the included manuscripts as cast reports and contains numerous references to various types of review studies. Putting all the information about the search criteria and methodology together left me quite confused about what the authors aimed to do and had achieved. All sections concerning the methodology need a rigorous rewrite to clarify what was actually done.

We thank the reviewer for this remark. We proceeded to a two-fold research strategy, “review” was a search criterion for the first part of this research, which was completed by a research focused on the main categories of pathologies. We excluded case reports in order to avoid special and rare cases that are not relevant to this review. However, some of the selected articles dealing with psychopathology concerns or presenting small reviews are illustrated by a case report.

>> Lastly, as a minor note, I think page 3 line 121 should contain a reference to figure 1, but now contains an error message, and the actual figure has “table 1” as a legend, which I don’t feel is appropriate.

                We thank the reviewer for this remark. References to figures and tables have been updated.

>> Readability and understandability for a broad audience: First, the description of the results for limbic encephalitis immediately starts with an overview of associated antibodies but does not entail a brief description of what it is (page 4, lines 128-129). This contrasts with other sections where some form of brief description is given. As I believe the readership of this journal may well be highly diverse, I would generally recommend giving more general descriptions of what the various disorders detail as the level of detail provided at present differs from disease to disease.

We thank the reviewer for this remark. In the description of the results for limbic encephalitis, we specified that it is an autoimmune brain inflammation: “Since the first description of limbic encephalitis in 1960, several antibodies responsible for these autoimmune brain inflammations have been discovered.” We have also added a table listing the main symptoms and paraclinical exams recommended for each pathology (Appendix B).

>> Second, from the sentence on page 10 lines 407-409, it is unclear what kind of systematic testing is required. Moreover, I assume that consent for testing is a more general criterion, not limited to this specific setting, while the reference to state laws suggests this may be US-specific. All in all, this sentence is hard to understand without returning to the cited article. Similarly, on page 11 lines 447-448 “CSF analysis is a key yes for the diagnosis of LNB, ich is defined as follows [ref 140]” cannot be understood without reading the cited articles. Overall, I recommend the authors to carefully check the manuscript for similar sentences and provide adequate information to understand the point they are trying to make without the need to read the cited articles.

                We thank the reviewer for this remark. Concerning HIV, we specified the systematic testing required as follows: “Systematic testing (detection of anti-HIV antibodies in serum) is recommended in the case of FEP but consent from each patient may be required, according to some state laws, before testing [127].” We consider important to point out that the legislation relating to consent to HIV testing may vary from state to state, as this does not only apply to the United States. For example, in Europe, French legislation requires patients to give their consent before an HIV test is carried out.
The diagnosis of LNB is based on the presence of several criteria, which are summarised in the article of Mygland and al. We modified the sentence as follows: “The CSF analysis is a key test for the diagnosis of LNB, which is based on several criteria defined in the article by Mygland and al. [141].”

>> Third, in section 3.6 genetic diseases (page 11, lines 450-485), unlike the previous sections, the recommendations for differential diagnosis are lacking. While even lay people will understand this will require some form of genetic screening, for consistency across the review as well as for clarity, I recommend the authors to either add a general statement on how genetic screening may be achieved to the introductory paragraph of this section or to describe the specific, preferred genetic test for each of the disorders.

                We thank the reviewer for this remark. We have added a general statement in the introductive part on genetic diseases: “The relationships between some genetic disorders and psychotic-like symptoms have been noted since the 1980’s [142]. Because of the multiplicity of possible gene alterations and the low prevalence of these diseases, however, a systematic genetic screening is not yet conceivable. However, a genetic screening should be discussed in case of clinical presentation evocative of a genetic disorder. In most of these diseases, patients suffer from several physical disorders and may present with an intellectual disability, dysmorphia, and resistance to antipsychotic treatments. The genetic syndromes reported in the literature are 22q11 deletion syndrome, Prader-Willi syndrome, C9orf72 mutations, Fragile X premutation disorders [143,144], BSC1L mutation [145], 3q29 recurrent deletion [146], 15q duplication syndrome [147] and genetic mitochondrial disorders [148]. The most frequently found syndromes in cases of FEP are described below.”

>> Finally, table 2 (page 13 line 575) summarizes the types of testing the authors recommend in section 4.4 exams (pages 12-14, lines 528-576), but no reference to this table is made in the main text.

We thank the reviewer for this remark. A reference to the table summarizing the recommended tests has been added to the conclusion.

Round 2

Reviewer 1 Report

Thank you very much for the opportunity of reviewing the manuscript. The authors have addressed all the comments.

Reviewer 2 Report

I would like to thank the author for clarifying the search strategy in their author response and I am happy to see the addition of Appendix B in regard to my question readability and understandability for a broader audience, I believe this appendix to be of great assistance. I'm also satisfied with the explanations and additions to the manuscript based on remarks for specific lines.